# Gender-Related Approach to Kidney Cancer Management: Moving Forward

**DOI:** 10.3390/ijms21093378

**Published:** 2020-05-10

**Authors:** Mariangela Mancini, Marialaura Righetto, Giovannella Baggio

**Affiliations:** 1Department of Surgical, Oncological and Gastroenterological Sciences, Urological Clinic, University of Padua, 35128 Padua, Italy; mariangela.mancini@unipd.it; 2Department of Molecular Medicine, School of Medicine and Surgery, University of Padua, 35128 Padua, Italy; giovannella.baggio@aopd.veneto.it

**Keywords:** sex and gender, kidney cancer, outcomes, renal cell carcinoma, hormones profile and inflammation, sex-specific disease biomarkers, drug response and resistance

## Abstract

Men are more frequently diagnosed with kidney cancer than women, with a more aggressive histology, larger tumors, a higher grade and stage, and worse oncological outcomes. Smoking habits and sex steroid hormones seem to have a possible role in explaining these gender disparities. Moreover, the expression of genes involved in tumor growth and immune response in kidney cancer varies between men and women, having an impact on the gender-related response to oncological therapy, such as anti-angiogenic drugs and immunotherapy. Recent advances have been made in our understanding of the molecular and genetic mechanisms involved in kidney cancer, which could partially explain the gender differences, and they are summarized in this paper. However, other key mechanisms, which fully clarify the striking clinical gender-related differences observed in kidney cancer, are not completely understood at present. We reviewed and summarized the most relevant publications about the relationship between gender and kidney cancer. Efforts should be made to progress in bench and clinical research on gender-related signatures and disparities, and their impact on the clinical management of kidney cancer.

## 1. Introduction

Renal cell carcinoma (RCC) represents 2–3% of all cancers, with the highest incidence in Western countries [1], and is responsible of 5% of cancers in men and 3% in women [2]. Over the last two decades, the incidence of RCC has increased worldwide by about 2%, with the lowest rates in Africa and Asia. In 2012, there were approximately 84,400 new cases of RCC and 34,700 RCC-related deaths in the European Union [1]. The incidence of RCC has been shown to have risen 2.1-fold between 1990 and 2013 [3], although there are currently no recommendations for a systematic screening for RCC. This could be the result of improved imaging techniques that increase incidental diagnosis of localized RCC [1].

Gender-related differences have been found to be associated with tumor characteristics, and with surgical, functional, and oncological outcomes of RCC. The most recent and clinically relevant findings on this topic are reviewed in this paper.

Taking into account such findings, when planning the management of kidney cancer cases, represents a novel and promising strategy, as we move forward and approach the era of personalized and precision medicine.

## 2. Materials and Methods

We performed a nonsystematic review of the literature through the electronic databases PubMed and Scopus using the following keywords: “kidney cancer”, “renal cell carcinoma”, “gender”, “female”, “male”, “incidence”, “outcomes”, “prognosis”, and “survival”. For the purpose of the study, we screened the most relevant articles published in the last 10 years, between January 2010 and January 2020. Some older studies have been included on the basis of their relevance. Selected articles were independently reviewed by the authors.

## 3. Epidemiology of RCC and Gender

The incidence of RCC is higher in males than in females. In a recent retrospective study by Gelfond et al. [4], men had a 1.85-fold higher risk of RCC than women. However, when compared with other solid tumors, RCC showed the lowest variability in the male/female sex ratio worldwide over the last 30 years [5].

RCC has a 2:1 male/female incidence ratio, which is stable by age over time, thus socio-cultural and health-related behaviors (i.e., cigarette smoking, hypertension, obesity) are not the only factors responsible for this disparity [5]. However, there is some data showing that males have a twofold higher incidence of RCC at 40–60 years of age as compared to women, and that this difference tends to progressively disappear at ≥70 years [6]. Moreover, females are significantly older than men at the time of diagnosis (3 years, median) [6]. These data show a possible protective role of female hormones, like oestrogens, at younger age.

Hypertension, which is more common in men than in women, has been recognized as a modifiable risk factor for RCC [7]. Men with hypertension have a 1.32-fold higher risk of RCC than women. However, the effects of hypertension on RCC development appear to be more severe in females [4].

Regarding other risk factors, cigarette smoking, a well-known carcinogen, is more frequent in men than in women. On the other hand, there is no significant difference in the effect of body mass index (BMI) on RCC gender-related incidence [4].

A recent retrospective study by Lotan et al. highlighted the presence of a population at high-risk of RCC that deserves a rational screening. This group included over 60-year-old males, current smokers, and the obese [8].

## 4. Gender-Related Histological Characteristics of Kidney Tumors

RCC includes several histopathological subtypes, defined in the 2016 World Health Organization (WHO) classification, with different cytogenetic and genetic features [9].

There are three main RCC types: clear cell (ccRCC), papillary (pRCC—type I and II), and chromophobe (chRCC) carcinoma; ccRCC is the most common (~70% of cases), pRCC accounts for 10–15% of cases, and chRCC for 5%.

Histological subtypes are linked to overall survival (OS) and cancer specific survival (CSS), with a better prognosis for the chRCC subtype (vs. pRCC vs. ccRCC) [10]. However, prognostic information provided by the subtype is lost when stratified to tumor stage [11]. On the contrary, patients with rare histologic variants of RCC, such as collecting duct and sarcomatoid variants, have poor survival even presenting with low stage disease [11].

Only a few studies have been published so far on gender-related distribution of histological features and oncological outcomes of RCC: the available data show that females tend to have a higher frequency of more favorable histopathological subtypes at the time of first diagnosis, and are less likely to present with pRCC than with chRCC. These pathological gender-related differences could also explain the more favorable outcome of RCC in women.

According to the multicentre Collaborative Research on Renal Neoplasms Association (CORONA) database [12], on 6234 patients (3751 males and 2483 females), females are diagnosed more frequently with ccRCC (82.4% vs. 78.6%; *p* < 0.001) and chRCC (5.2% vs. 3.5%; *p* = 0.001), while men more often with pRCC (9.7% vs. 15.2%; *p* < 0.001). These data have been confirmed in a recent large retrospective study on 1532 patients submitted to partial or radical nephrectomy for RCC [13]. Compared with ccRCC cases, patients with pRCC were significantly less likely to be female (odds ratio (OR): 0.60; 95% confidence interval (CI): 0.43–0.83), while patients with chRCC were significantly more likely to be female (OR: 2.32; 95% CI: 1.44–3.74). Despite the lower number of RCCs in Asia, a recent Japanese retrospective study [14] on 5265 patients with RCC (72.6% male, 27.4% female) confirmed the different distribution of histological subtypes between men and women. While the prevalence of ccRCC/“others” subtypes was similar between genders, pRCC was more prevalent among men, as compared to women (4.6% vs. 2.8%; *p* = 0.004), and chRCC was less prevalent among men, as compared to women (1.6% vs. 4.8%; *p* < 0.001). Overall, women had a 0.6-fold lower prevalence of pRCC and a 3.2-fold higher prevalence of chRCC.

An important issue for surgeons involved in RCC surgical treatment is the necessity to reduce the incidence of benign lesions in the final histological report of small renal masses submitted to nephron sparing surgery (NSS) (pT1). Taking into account the surgical and functional complications of NSS, it is mandatory to try to reduce the incidence of benign final histological reports, which could have been spared a dangerous surgical procedure. The prevalence of benign histological findings after NSS for small renal masses was recently reported to be as high as 8% to 30% [15]. Several studies have focused on the relationship between gender and the rate of final benign histological reports, showing that female sex and younger age are the main predictive factors for it. This implies that women are submitted to NSS for benign lesions more often than men, which could therefore have been avoided. In a large cohort study on 18,060 patients submitted to NSS (58.9% males, 41.1% females) [16], 5588 (30.9%) had a benign histological diagnosis at final pathology. In this group of benign lesions, female gender was prominent, as compared with the malignant tumor group (48.9% vs. 37.6%, respectively), with women having a 0.62-fold increased risk of benign histological diagnosis than men (OR: 0.62, 95% CI: 0.58–0.66; *p* < 0.001).

Another study by Mauerman et al. [17] confirmed that female gender is an independent predictor of benign histology after renal surgery. Women showed a >2-fold higher chance of benign pathological findings, as compared to men. In addition, women presented with a diagnosis of angiomyolipoma more frequently than men (72% vs. 28%), while oncocytoma was more frequent in men (59% vs. 41%). Additionally, men with benign histological findings were significantly older, with a higher BMI and Charlson comorbidity score, lower Eastern Cooperative Oncology Group (ECOG) performance status, and smaller tumors than women. These data deserve very careful consideration when planning NSS in cases of a woman with a small renal mass, given that women have a higher risk of this small mass ending up being diagnosed as a benign lesion on final pathology.

## 5. The Role of Sex Hormones

The unbalanced male–female ratio of RCC incidence suggests that there is an influence of sex hormones and their receptors on RCC development and progression, as already reported in bladder cancer [18,19].

Noh et al. [20] recently showed that a higher expression of androgen receptor (AR) is associated with cases of ccRCCs with a higher nuclear grade, shorter OS, CSS, and recurrence-free survival (RFS), regardless of gender. Zhu et al. [21] confirmed that there is no statistically significant difference in the rate of AR-positive RCC in men and women, showing that women with RCC have tumors expressing the AR in the same fashion as men, and that this receptor’s expression is associated with worse outcomes and more aggressive features.

Additionally, Chen et al. [22] showed that higher AR expression has a positive role in promoting RCC growth and proliferation. Moreover, the authors identified a specific micro-RNA (miRNA-145) related to RCC growth control and suppression, and found that the AR could downregulate miRNA-145 expression to enhance RCC proliferation. This finding implies that targeted molecular therapy and anti-androgen therapy could help to better suppress RCC growth and progression.

On the other hand, the oestrogen receptor-β (ERβ) is more expressed in RCC cells than in breast cancer cells [23], and might have a protective role in RCC growth, acting as a tumor suppressor. Oestrogen stimulation and oestrogen receptor-β (ERβ) activation result in the inhibition of proliferation and in the induction of apoptosis in RCC, and this fails after ERβ downregulation. Moreover, the role of ERβ as a suppressor of RCC growth offers a clear possible explanation for the gender-related difference in RCC incidence, which is higher in men than in women.

These data seem to show that sex hormone receptors could represent druggable targets for RCC treatment, individualizing treatment on the basis of expression of the receptors in the single tumors.

Moreover, the presence of hormonal receptors in RCC cells could suggest the hypothesis that female reproductive and menstrual factors, including hormonal therapy, have a role in RCC aetiopathogenesis and growth modulation [24]. A prospective cohort study on 229 women with RCC (in a cohort of 106,036 women) by Setiawan et al. [25] showed that later age at menarche (≥15 years) had a 42% elevation in risk of RCC, but without reaching statistical significance (*p* = 0.13). Later age at first delivery (≥26 years) had a lower risk of RCC than those who gave birth at <20 years; while women with ≥5 children had a 1.31-fold higher risk of RCC, compared to those with one to two children. However, none of these factors were found to be statistically significant. Moreover, neither oral contraceptive nor hormonal therapy use was significantly associated with RCC risk, and no clear relationship was found between menopause age/history of hysterectomy or oophorectomy and RCC development [24]. On the other hand, a meta-analysis by Liu et al. [26], involving 4206 RCC cases, suggested that oral contraceptive use (especially long-term use) could reduce the risk of RCC (relative risk (RR): 0.89, 95% CI: 0.82–0.98), and that cancer risk decreased by 2% every year of oral contraceptive use. Finally, regarding hysterectomy and RCC risk, a systematic review and meta-analysis by Karami et al. [27] showed that women who underwent a hysterectomy had a 30% increased relative risk of RCC (RR: 1.29, 95% CI: 1.16–1.43), regardless of age at hysterectomy. This last finding is interesting as it demonstrates the possible protective role of preservation of the uterus on the incidence of kidney cancer, and deserves further investigation.

## 6. Genetic Factors

Genetic features are different between men and women with RCC. In particular, ccRCC is the subtype with a more evident molecular heterogeneity, even between sexes. Moreover, two dominant subtypes of ccRCC have been described, ccA and ccB, with different gene expression and different cancer specific survival (8.6 yrs vs. 2 yrs, respectively; *p* = 0.003) [28].

A meta-analysis from 2012 of six large ccRCC databases (366 autosomal genes) [28] analyzed the meta-arrays based on gender. This study showed a distinct metabolic pattern of ccRCCs between men and women for approximately 89% of genes, and several of these genes encode proteins that have a role in important biological pathways. Specifically, ccRCC from males had an overexpression of immune or inflammatory genes, while ccRCC from females overexpressed genes involved in the catabolic process. These findings underlined the importance of the immune system and inflammation as potential targets for ccRCC treatment in men, more than in women. Moreover, the overexpression of genes related to the catabolic process of female tumors suggests a potential therapeutic approach by promoting a metabolic switch in these tumors.

A recent sex-specific genome-wide association analysis [29] on a dataset of 13,230 patients (8193 men, 5087 women) with RCC underlined a gender-related association for four single-nucleotide polymorphisms (SNPs) on genes *DPF3*, *EPAS1*, *SAMD5*, and *BTBD11*. *DPF3* is a chromatin-remodeling complex associated gene related to a greater risk of RCC in women. *EPAS1* (HIF2-α; hypoxia-inducible factor 2-α) has a stronger association to RCC in males (OR: 1.18). The last two genes are associated with a risk for men, without strong evidence of association for women.

The X-chromosome encoded genes are more prevalent in male-associated tumors. Ricketts et al. [30] performed an analysis of three large ccRCC mutation sequencing datasets, and found, in both sexes, mutations of *KDM5C*, histone demethylase, or *BAP1*, and the gene coding for a deubiquitinase (both being X-chromosome encoded genes). Mutation of *BAP1* was associated with a poorer overall survival (*p* = 0.0039) in all, and in females (*p* = 0.0021), but not in males. Mutation in *KDM5C* did not reduce patients’ survival in total or by gender.

## 7. Impact of Gender on Prognosis

Male gender is associated with worse RCC clinical features and prognosis, and several studies investigated the prognostic impact of gender in RCC.

RCC mortality data were collected retrospectively from global vital registries by Dy et al. [3]. The data showed that smoking-related RCC decreased in trend (from 28.8% in 1990 to 22.4% in 2013, although remaining the major risk factor globally), while obesity-related deaths rose, especially in women. Deaths from RCC increased across all genders and age groups, by 1.1% per year, but age-standardized death rates remained stable in both genders (the 11.3% decrease mortality in women balanced the 15% increase in mortality in men).

A large retrospective study based on an Australian cancer registry [31] showed, as regard to RCC, a higher mortality rate for men than for women.

In a cohort study by Bhindi et al. [32], on 2650 patients who underwent partial or radical nephrectomy for pT1-2 pNx/0 M0 RCC, the percentages of malignant/aggressive histology were higher for men. Moreover, the likelihood of malignancy for a given tumor size in men was comparable to tumors twice as large in women.

In the Surveillance Epidemiology and End Results (SEER) database [33] on 35,336 patients with RCC (63.1% male), tumors in females were significantly smaller as compared to men (5.9 cm vs. 6.1 cm), and with a lower pathological grade. Males, on the other hand, had a higher incidence of regional or metastatic spread of RCC. Cancer-specific survival and overall survival were superior for females (HR = 0.85 and HR = 0.86; *p* < 0.0001; respectively) as compared to males. In the multivariate analysis, however, females confirmed the OS advantage, but did not present a better CSS.

In the CORONA database [12], male gender was associated with a more unfavorable grading than female (grade 3–4 tumors; 22.8% vs. 17.8%) and males presented a higher incidence of distant metastasis (7.6% vs. 5.8 %). However, unlike the study on the SEER database, maximum tumor diameter and pTN stages were similar in both sexes [12]. Women had significantly better outcomes than men, with higher CSS (HR: 0.75; *p* < 0.001) and OS (HR: 0.80; *p* < 0.001), but, once again, not in the multivariate analysis.

In both studies [12,33], the gender-related difference in tumor stage at presentation (with women being diagnosed at an earlier stage than men) was associated with a more frequent incidental diagnosis of localized RCC in women. Moreover, in another study, women with Stage 4 RCC were significantly less numerous than men (42% vs. 35%; *p* < 0.05), while men tended to have a more advanced stage of disease, with 56% being Stage 2–4, compared to 29% of women [34].

The multivariate analysis of a study cohort on 57,700 patients with localized disease (T1-2 N0 M0) from the SEER database [35] presented male gender as a strong predictor of T2 stage RCC (OR: 1.12, CI 1.07–1.18; *p* < 0.001) and grade G3/G4 (OR: 1.35, CI 1.30–1.41; *p* < 0.001). Moreover, unmarried male patients were significantly associated with higher cancer-specific mortality (CSM).

The gender-related difference in biological aggressiveness remains unknown, being potentially linked to the expression of different sexual hormones. In this regard, there is no significant gender difference in recurrence rate in patients with N0M0 ccRCC after nephrectomy, even if the prognosis is more favorable in women, after recurrence.

Another important aspect to consider when comparing different treatment outcomes between genders is age: for localized RCC, the data report no differences in disease-specific mortality (DSM) between males and females, limited to patients >59 years of age.

In a large multicentre RCC cohort study on 5654 patients (67% men, 33% women) [36], women had a 19% improved CSS as compared to men (HR: 0.81; 95%CI 0.73–0.90; *p* < 0.001). This gender-related survival advantage was present only in women aged <59 years and not for women older than 59 years (*p* = 0.248). Age was not related to prognosis in men (*p* = 0.396).

Not all studies present a higher cancer-specific survival and overall survival in women rather than in men. Specifically, Zaitsu et al. [14] analyzed a cohort of 5265 patients with RCC and equally distributed grades and stages (high grade: ~10%, *p* = 0.85; late stage: ~26%, *p* = 0.40). In the survival analysis, the five-year OS rate was 72% for both males and females, without statistically significant differences between genders, even after stratifying by histological subtypes. The authors stated that this fact could be due to better prognostic factors in the Western female population (smaller tumor size, low pathological grade, early stage) as compared to men.

## 8. Surgery and Surgical Outcomes

The complexity of nephron-sparing surgery (NSS) for RCC could be predicted, in a presurgical setting, by nephrometry scores: the higher the score, the more complex the NSS.

A prospective cohort of 99 patients evaluated the association between obesity and RCC characteristics detected in the R.E.N.A.L. nephrometry score [37]. The authors found that higher waist circumference and male gender were significantly associated with increased R.E.N.A.L. score; particularly, males had an average R.E.N.A.L score 1.11 units higher than females.

A recent retrospective study by Ito et al. on 111 RCCs cT1aN0M0 [38] showed that males had a higher retroperitoneal fat tissue thickness (determined by computed tomography scan) than females, and that, as a consequence, the operative time during retroperitoneal laparoscopic NSS was significantly longer in male patients than in female patients (181.8 ± 34.9 min in males, 150.7 ± 24.7 min in females; *p* < 0.001). Overall, the operative time of retroperitoneal laparoscopic NSS was significantly correlated not only with gender, but also with maximum tumor diameter, and retroperitoneal fat tissue thickness. In the transperitoneal laparoscopic NSS, no difference was observed in the operative time between genders.

This could be related to the management of the fat tissue surrounding the kidney being different and more difficult, as it is more compact in males.

Patients’ selection for NSS or non-NSS for small renal masses has not been based, so far, on patients’ gender. However, some studies reported different treatment options for males and females.

A recent study on the Surveillance Epidemiology and End Results (SEER) database [35] reported that male gender predicted higher risk for nonsurgical treatment (OR: 1.23, CI 1.13–1.33; *p* < 0.001) in a population-based study on T1-2N0M0 RCC.

On the other hand, the CORONA database analysis (RCCs all submitted to NSS) [12] showed that males were significantly younger at diagnosis (62 vs. 64.8 years) and underwent NSS (despite a similar median T-stage) significantly more often (21.7% vs. 18.4 %; *p* = 0.002), compared to females. This last piece of evidence could be related to the more advanced age at diagnosis in females.

Tan et al. [39] in a cohort of 15,871 patients (T1N0M0) from the SEER database reported that the likelihood of NSS was significantly associated with younger age, higher socioeconomic position, smaller tumor size, and male sex.

Another retrospective study by O’Malley et al. [40] analyzed 386 patients (156 women, 30 men) with a localized, ≤cT2 RCC, and median R.E.N.A.L. score of 8 for both sexes.

Women had a more frequent history of previous abdominal surgery (55.8% vs. 31.7%), a significantly lower preoperative estimated glomerular filtration rate (eGFR), and a significantly lower Charlson index than males (38% vs. 49%; *p* = 0.02). At logistic regression, women were 2.46 times more likely to undergo radical nephrectomy, as compared to men (OR: 2.46, 95% CI: 1.24–4.87; *p* < 0.01). No clear reason for this gender-related difference was found by authors.

Gender-related differences can influence the incidence of postoperative complications, as shown in a Canadian population-based retrospective study on 20,286 radical and 4292 partial nephrectomies [41]: in-hospital morbidity after renal surgery was significantly lower for women. Women had a lower overall complication rate (OR: 0.94, 95% CI: 0.88–0.99), and a lower in-hospital mortality rate (OR: 0.64, 95% CI: 0.49–0.83); conversely, men showed significantly higher rates of wound, nephrectomy-related, and medical complications, and these complications became more relevant with increasing age. Moreover, women showed fewer complications than men after radical nephrectomy, but there were no differences between genders in complication rates after NSS.

## 9. Other Factors

Regarding long-term functional outcomes, a recent retrospective single-center analysis on 402 patients treated with NSS [42] showed that statistically significant variables associated with acute kidney injury and lower eGFR after NSS were right-side tumors, male sex (*p* = 0.01), hypertension, and a history of nephrolithiasis. However, in another recent subgroup analysis of a EORTC 30904 Phase 3 randomized trial conducted in patients with a small (≤5 cm) renal mass (two-thirds of patients being male), the variables associated with lower eGFR included radical nephrectomy (vs. NSS), older age, basal chronic disease, and female sex [43].

Concerning long-term oncological outcomes, a study based on the CORONA database [12] showed a strong significant correlation between female gender and improved OS and disease-free survival (DFS). In particular, the DFS at 5 and 10 years was 84% and 75% in males and 89% and 80% in females, respectively (*p* < 0.001). The corresponding overall survival rates were 76% and 61% in males and 81% and 65% in females, respectively (*p* < 0.001). Female patients had a 25% decreased risk of cancer-specific mortality (CSM), compared with men (HR: 0.75, *p* < 0.001). However, no significant gender difference in OS and DFS was found in patients with M0 RCC undergoing NSS.

In a retrospective study on 313 patients who underwent cytoreductive nephrectomy for metastatic RCC [44], female sex was significantly associated with CSS (HR: 1.9), together with older age (≥75 years), constitutional symptoms (i.e., rash, weight loss, fatigue, anorexia), radiographic lymphadenopathy, and tumor thrombus in the inferior vena cava.

Finally, the unmarried status (separated/divorced, widowed, or single) is an adverse risk for oncological outcomes and CSS both for metastatic and nonmetastatic RCC. Recently, a SEER database analysis on 6975 patients with metastatic ccRCC was performed [45].

Overall, unmarried men were at higher risk of not profiting from cytoreductive nephrectomy (OR: 0.54), metastasectomy (OR: 0.70), and systemic therapy (OR: 0.70), while women with an unmarried status were at higher risk of not profiting from cytoreductive nephrectomy or systemic therapy. Furthermore, unmarried men had a higher CSM than unmarried women.

## 10. Systemic Therapy

Approved targeted therapy agents in advanced and metastatic RCC include bevacizumab (a monoclonal antibody that prevents vascular endothelial growth factor (VEGF)-A binding to its receptor), tyrosine kinase inhibitors (TKIs, that directly inhibit the VEGF receptor), inhibitors of the mammalian target of rapamycin (mTOR) complex (i.e., temsirolimus, everolimus), and immune checkpoint inhibitors against programmed cell death protein-1 (PD-1) (nivolumab) and against cytotoxic T-lymphocyte-associated protein 4 (CTLA-4) (ipilimumab). The selection of patients with metastatic ccRCC who might benefit from these treatments remains challenging.

Checkpoint inhibitors block the immune inhibitory signals of tumor cells, and as a consequence, stimulate the immune response. Overall, male solid tumors are more antigenic than female tumors, so that checkpoint inhibitors are more effective in males per se than in females [46]. For these reasons, different therapeutic strategies have been proposed: the immune environment could be enriched in males, whereas tumor antigenicity could be enriched in females (while therapeutic strategies that improve immune response are less effective in women than in men) [47].

Recent meta-analyses on gender-related differences in efficacy of checkpoint inhibitors in metastatic RCC patients reported contradictory data.

Motzer et al. [48] recently showed that patients with advanced RCC who had received previous antiangiogenic therapy had longer survival with nivolumab than with everolimus.

In a subgroup analysis of OS, males had a clear survival advantage with nivolumab (HR: 0.73; 95% CI: 0.58–0.92), while in females there was a survival advantage, but it was not as significant (HR: 0.84; 95% CI: 0.57–1.24). Another multicenter, randomized, open-label, Phase 3 trial [49] comparing avelumab plus axitinib (*n* = 442) with sunitinib (*n* = 444), confirmed, in the subgroup analysis of progression-free survival (PFS), that both gender had a survival advantage with avelumab plus axitinib, but, once again, males had a clearer survival advantage for disease progression or death (HR: 0.56; 95% CI: 0.42–0.75) than females (HR: 0.90; 95% CI: 0.55–1.47).

A recent systematic review and meta-analysis by Conforti et al. [50] on 11,351 patients with advanced or metastatic solid cancers showed that male patients had a significantly reduced risk of death when treated with immune checkpoint inhibitors, as compared to females.

In response to this study, Graham et al. [51] analyzed outcomes of patients with metastatic RCC (mRCC) treated with the immune checkpoint inhibitor (nivolumab and everolimus) using the International Metastatic Renal Cell Carcinoma Database Consortium (IMDC). In the setting of mRCC, the authors concluded that there is no evidence to recommend gender-related bias for immunotherapy, even if the magnitude of efficacy of nivolumab was double for males (12.2 months of OS improvement for males, 5.8 months of OS improvement for women). Unfortunately, despite the higher efficacy of nivolumab in males, the difference between genders was not strong enough to make clear clinical recommendations based on gender.

The observed heterogeneity between sexes could be explained by biological and physiological differences.

More recently, a meta-analysis by Hassler et al. [52] included four randomized controlled trials comprising a total of 3664 patients. Immunotherapy gave an OS and PFS advantage both in men and women, and no statistically significant difference between genders was observed (OS advantage: HR: 0.69 for men, HR: 0.62 for women; PFS advantage: HR: 0.7 for men, HR: 0.68 for women).

## 11. Differences in Psychological Distress

A Canadian retrospective analysis on 495 nonmetastatic RCC patients [53] showed that, despite having similar demographic, clinical, and treatment characteristics to men, female patients revealed higher psychological and physical distress scores, after diagnosis, biopsy, and surgery. Furthermore, younger age of the female patients was associated with higher levels of distress.

In another recent survey on 450 patients with RCC (nonmetastatic RCC: 74%; disease recurrence: 61%; ccRR: 76%) [54], distress was significantly associated with female gender, younger age, non-ccRCC, and the presence of recurrence.

Both studies showed, in almost 1000 cases, that there is a strong association between female gender and psychological distress linked to kidney cancer. Reasons for these data are not fully understood and require further investigation.

## 12. Conclusive Remarks

Gender seems to influence incidence, histology, surgical treatment (including its complications), response to medical therapy, psychological consequences, and long-term functional and oncological outcomes in RCC (Table 1). Men are at a higher risk of developing RCC and usually have a more aggressive disease at the time of diagnosis. Females present more often with favorable histological RCC subtypes, have better oncological outcomes than males, and are more likely to present a benign histology after NSS. On the contrary, in the metastatic RCC setting, female gender seems to be linked to a worse response to therapy and shorter survival. However, we must underline that most of the data come from studies published in the developed world; it is possible to hypothesize that in countries where women have limited access to medical care (including ultrasonography or computed tomography), they would be more likely diagnosed with RCC in a more advanced stage, as compared to men. This could be particularly true in countries with limited freedom and civil rights for the female population, considering that RCC is often a totally asymptomatic disease.

At present, the biological, genetic, and molecular pathways that explain the gender-related differences in RCC are not fully understood. A better clarification of gender-related mechanisms, a process that is currently moving forward, could lead to the possibility of including gender factors in risk-predictive nomograms. Efforts should be made to accelerate progress in bench and clinical research on the gender impact in kidney cancer, in order to move towards personalized gender-oriented treatment options for the patients.

## Figures and Tables

**Table 1 ijms-21-03378-t001:** Gender-related differences in renal cell carcinoma (RCC).

Factors	Males	Females
Epidemiology	40–60 years: double incidence than females [6] ≥70 years: similar incidence than females [6] Younger at diagnosis [6] If hypertension: 1.32-fold higher risk of RCC over women [7]	40–60 years: half incidence than males [6] ≥70 years: similar incidence than males [6] Older at diagnosis [6] If hypertension: more severe disease [4]
Histology	Prevalence of ccRCC similar to women [12,15] More pRCC [12,15] Larger tumors, higher grade, higher stage [12,15]	Prevalence of ccRCC similar to men [12,15] More chRCC [12,15] Smaller tumors, lower grade, lower stage [12,15] >Twofold higher diagnosis of benign histological reports in resected small renal masses [12,15]
Sex hormones	Androgens promote tumor growth [22]	Androgens promote tumor growth [22] Oestrogen could have a protective role [23] Later age at first delivery: protective role [25] Increased parity: increased risk [25] Isterectomy: increased risk [27] Oral contraceptive: reduced risk (controversial) [24,26]
Genetics	Genes of immune or inflammatory response [28] 2p21 (EPAS1 gene) + 6q24.3 [29] (SAMD5 gene) + 7p22.3 [29] (BTBD11 gene): higher risk [29]	Genes involved in the catabolic process [28] 14q24.2 (DPF3 gene): high risk [29]
Systemic therapy	Men could have a higher survival advantage with TKIs and/or CPIs than females [48]	
Oncological outcomes	Poorer outcomes [12,44]	Better outcomes [12,44]
Surgical outcomes	More NSS [12,39] Higher complications rate [41]	More RN [12,39] Lower complications rate [41]

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
