# Peer review of "Gender-Related Approach to Kidney Cancer Management: Moving Forward"

_ijms, 2020, doi:10.3390/ijms21093378_

Round 1

Reviewer 1 Report

Dear Authors,

I would like congratulate you great effort that you done to writing the review manuscript entitled “Gender-Related Approach to Kidney Cancer Management: Moving Forward”.
The subject of the study was to explore the sex-dependant RCC based on available publications describing clinical cohort studies, between January 2010 and January 2020. Authors performed a non-systematic review through the PubMed and Scopus using the keywords such as: ‘kidney cancer’, ‘renal cell carcinoma’, ‘gender’, ‘female’, ‘male’, ‘gender differences’, ‘disparities’, ‘incidence’, ‘outcomes’, ‘prognosis’ and ‘survival’.

Authors concluded that: “Gender seems to influence incidence, histology, surgical treatment, medical therapy and  outcomes in RCC. Men are at a higher risk of developing RCC and usually have a more aggressive disease at the time of diagnosis. Females present more often with favorable histological RCC subtypes, have better oncological outcomes than males, and are more likely to present a benign histology after NSS. On the contrary, in the metastatic RCC setting, female gender seems to be linked to worsen response to therapy and shorter survival”. And suggested that: “A better clarification of gender-related mechanisms could lead to the possibility to include gender in risk-predictive nomograms. Efforts should be made to make progress in bench and clinical research on gender impact in kidney cancer, in order to provide a personalized gender-oriented treatment plan to the patients”.

In my opinion, the manuscript is written coherently, consistently and factually, and is a source of valuable information for not only clinicians but also for scientists that studied this field of subject.

Notwithstanding, I have some comments, see below.

General comments:

  • Please expand all shortcuts of database that you describe/comment
  • Please harmonize the use of either dots or commas after thousands in cypher/numbers (i.e. Page 1: In 2012, there were approximately 84,400 new cases of RCC and 34,700 RCC-related deaths in the European Union1. Or: Page 2: According to the multicentre Collaborative Research on Renal Neoplasms Association (CORONA) database 31 (ßin correct number of reference it should be 12), on 234 patients (3.751 males and 2.483 females), females are diagnosed more….or without any dots/commas like on Page 4: A recent sex-specific genome-wide association analysis on a dataset of 13.230 patients (8193 men, 5087 women) with…..
  • Please combine individual paragraph of the chapters (subsections) into longer paragraphs/parts of the text
  • Please add to the manuscript the list of used abbreviations
  • Please correct the citation: please use everywhere superscripts, and the superscripts should be before (not after) the dot which ending the specific sentence.
  • Please also correct the References according to Guide for Authors (among others page numbering: i.e.: …. Curr Hypertens Rep 2013;15:321-330…..; …. J Clin Oncol 2005;23(12):2763-2771…
  • Please add the citing references in to the Table 1.
  • Cannot chapter Outcomes be combined with 9. Clinical outcomes, or can chapter 7 be named differently?
  • What does HR, OR Cl mean? Please explain.

Detailed comments:

Page 2.

  1. Materials and Methods: …… the following keywords: ‘kidney cancer’, ‘renal cell carcinoma’, ‘gender’, ‘female’, ‘male’, ‘gender differences’, ‘disparities’, ‘incidence’, ‘outcomes’, ‘prognosis’ and ‘survival’. ß in my opinion to many keywords were used to find the appropriate publications
  2. Epidemiology of RCC and gender

….In a recent retrospective study by Gelfond J et al.4, men had 1.85-fold higher risk of RCC over women at Cox (ß please give information what does COX mean?) proportional hazards…

….Moreover, females are significantly older than men at the  time of diagnosis (3 years, median) 6.6; and this data shows a possible protective role of female hormones,  like oestrogens, at younger age.

Hypertension, which is more common in men than in women, has been recognized as a modifiable risk factor for RCC.7 Men with hypertension have a 1.32-fold higher risk of RCC over women. However, the effects of hypertension on RCC development appear to be more severe in females4.4 Another large epidemiological study3 3 showed that men had the greatest increase of age- standardized incidence rates over 20 years, worldwide, as compared to women.

Regarding other risk factors, cigarette smoking, a well-known carcinogen, is more frequent in men than in women. On the other hand, there is no significant difference in the effect of body mass index (BMI) on RCC gender-related incidence4.4   

Another large epidemiological study3 3 showed that men had the greatest increase of age- standardized incidence rates over 20 years, worldwide, as compared to women.

A recent retrospective study by Lotan Y et al 8, highlighted the presence of a population at high- risk of RCC that deserves a rational screening. This group included over 60-year old males, current  smokers and obese.8

  1. Gender-related histological characteristics of kidney tumours

RCC includes several histopathological subtypes, defined in the 2016 by World Health Organization (WHO) classification, with different cytogenetic and genetic features9.9

According to the multicentre Collaborative Research on Renal Neoplasms Association (CORONA) database 31 (ßin correct number of reference it should be 12),

Compared with ccRCC cases, patients with pRCC were significantly less likely female (OR ß what does it mean? 0.60; 95% CI ß what does it mean? 0.43–0.83), while…

Page 3:

…A recent Japanese (perhaps in this sentence should be emphasized once again that in Asia there is a lower percentage of RCC incidence) retrospective study on 5.265 patients with RCC….and please remove at the end of this paragraph bracket )

….Additionally, men with benign histological findings were significantly older, with higher BMI and Charlson comorbidity score, lower ECOG (Eastern Cooperative Oncology Group) performance status,

  1. The role of sex hormones

Page 4:

… Moreover, the presence of hormonal receptors in RCC cells could suggest the hypothesis that female reproductive and menstrual factors, including hormonal therapy, have a role in RCC aetiopathogenesis and growth modulation. ß please add references

….. Moreover, neither oral contraceptive nor hormonal therapy use was significantly associated with RCC risk, and no clear relationship was found between menopause age/history of hysterectomy or oophorectomy and RCC development. ß please add references

  1. Genetic factors

….This study showed a distinct metabolic pattern of ccRCCs between men and women for approximately 89% of genes, and several of these genes encode code for proteins that….

…. related to a greater risk of RCC in women. EPAS1 (HIF2-alfa; hypoxia-inducible factor 2- alpha) has a stronger….

…. Mutations on tThe X chromosome encoded genes are more prevalent in male-associated tumours.

  1. Outcomes

Page 5:

…. Deaths from RCC increased across all genders and age groups, by 1.1% per year…

…. CSS Cancer specific survival and OS overall survival were superior for females (HR=0.85 and HR=0.86; p<0.0001; respectively) as compared to males.

In the CORONA database31, male gender was associated with a more unfavourable grading than female (grade 3-4 tumours; 22.8 vs 17.8%) and males presented a higher incidence of distant metastasis (7.6 vs 5.8 %). However, unlike the study on the SEER database, maximum tumour diameter and pTN stages were similar in both sexes31. Women had significantly better outcomes than men, with higher CSS (HR=0.75; p<0.001) and OS (HR=0.80; p<0.001), but, one again, not in multivariate analysis. In both studies (ß is it possible to write like that, because in this sentence you cite another publication), the gender-related difference in tumour stage at presentation (with women to be diagnosed at an earlier stage than men) had been associated to a more frequent incidental diagnosis of localized RCC in women. Women with stage IV RCC are significantly less numerous than men (42% vs 35%; p<0.05) while men tend to have a more advanced stage of disease, with 56% being stage II-IV, compared to 29% of women32.32 ***

The gender-related difference in tumour grade (ßI would suggest remove tumor grade because in the paragraph above you writen about the differentiation of the tumor scale depending on gender) and biological aggressiveness remains unknown, being potentially linked to different sexual hormones expression

Another important aspect to consider when comparing different treatment outcomes between genders is age. More in detail, for localized RCC, data report no differences in disease-specific mortality (DSM) between males and females, limited to patients > 59 years of age. ß combine together this two paragraphs à In a large multicentre RCC cohort study on 5,654 patients (67% men, 33% women)33 women had a 19% improved CSS as compared to men (HR=0.81; 95%CI 0.73-0.90; p<0.001). This gender-related survival advantage was present only in women aged < 59 years and not for women older than 59 years (p=0.248). Age was not related to prognosis in men (p=0.396).

***The multivariate analysis of a study cohort on 57,700 patients with localized disease (T1-2 N0 M0) from the SEER database34, presented male gender as a strong predictor of T2 stage RCC (OR=1.12, CI 1.07–1.18; p<0.001) and grade G3/G4 (OR=1.35, CI 1.30–1.41; p<0.001). Moreover, unmarried male patients were significantly associated with higher cancer-specific mortality (CSM). ß this paragraph I suggest put in the other place within the text, (***)

Not all studies present a higher CSS cancer specific survival and OS overall survival in women rather than in men. ß combine together this two paragraphs à Zaitsu M et al.13 analyzed a cohort of 5,265 patients with RCC and equally distributed grades….

Page 6:

  1. Surgery, surgical and oncological outcomes

A recent retrospective study by Ito H et al.36 on 111 RCCs cT1aN0M036 showed that males had a higher retroperitoneal fat tissue thickness (determined by CT computer tomography scan) than females, and that, as a consequence, the operative time during retroperitoneal laparoscopic NSS was significantly longer in male patients than in female patients (181.8 ± 34.9 min in males, 150.7 ± 24.7 min in females; p<0.001). Overall, the operative time of retroperitoneal laparoscopic NSS was significantly correlated not only with gender, but also with maximum tumour diameter, and retroperitoneal fat tissue thickness. In the transperitoneal laparoscopic NSS, no difference was observed in the operative time between genders????. ßthe opposite information has already been said in the previous sentence

A recent study on Surveillance Epidemiology and End Results (SEER) database, reported …..

Another retrospective study by O’Malley RL et al.38 analysed 386 patients (156 women, 30 men) with localized, ≤ cT2 RCC, and median R.E.N.A.L. score 8 for both sexes.  Women had a more frequent history of previous abdominal surgery (55.8% vs 31.7%), a significantly lower preoperative eGFR (ß does it mean - glomerular filtration rate?), and a significantly lower Charlson index than males (38% vs 49%; p=0.02).  At logistic regression, women were 2.46 times more likely to undergo radical nephrectomy, as compared to men (OR=2.46, 95%IC 1.24-4.87; p<0.01). No clear reason for this gender-related difference was found by authors. Dear Authors, at the end I would like congratulate you conception of the studied very important information concerning kidney transplantation and social awareness about this life-saving procedure, and I suggests either re-written manuscript or writing a review paper. ß maybe this paragraph better suits to this above (lines 249-252) that concern R.E.N.A.L.

Gender-related differences can influence the incidence of post-operative complications. ß combine together this two paragraphs à In a Canadian population-based…

  1. Clinical outcomes maybe please consider to combine this chapter with 7. Outcomes

Page 7:

  1. Systemic therapy

… In the setting of mRCC, the authors concluded that there is no evidence to recommend gender-related bias for immunotherapy, even if the magnitude of efficacy of nivolumab was double for males (12.2 mo ß does it mean months of OS improvement for males, 5.8 mo of OS improvement for women)….

Dear Authors, at the end I would like congratulate you conception of the studied very important information concerning gender-dependent RCC occurrence, progression, prognosis and treatment, and I hope that after small correction the manuscript will be publish in International Journal of Molecular Sciences.

Best Regards,

Reviewer

Author Response

Dear reviewer,

we have made all the revisions required, using for clarity the software “Track Changes” (Microsoft Word).

The changes are listed below:

  1. We have expanded the shortcuts of the databases that we describe (line 164, line 303).
  2. We have harmonized the use of dots or commas in numbers (using always dots).
  3. We have corrected the number of reference 31, changing it into 12.
  4. We have combined individual paragraphs into longer parts of the text, as required.
  5. We have added a list of the used abbreviations.
  6. All the superscripts have been moved before the dot (not after).
  7. The references have been corrected according to the instructions in the Guide for Authors.
  8. We have added the references to Table 1
  9. We have redefined the chapters regarding the Outcomes, as required.
  10. We reduced the number of keywords, eliminating two of them
  11. We corrected the typos and wrong word spelling throughout the text.
  12. All the lines indicated as superfluous have been removed.
  13. We added the suggested sentences in order to make the text more readable.
  14. We added the requested references where they were missing.
  15. Regarding the surgical technique paragraph, in the text there is a sentence that refers to retroperitoneal technique, and another that refers to transperitoneal surgery. The two sentences refer to two different operations, so, the text can be left as it is.
  16. We combined the requested paragraphs and moved parts of the text as required.
  17. We have clarified the requested parts in the Systemic Therapy paragraph.

Reviewer 2 Report

The authors reviewed relevant articles about the relationship between gender and kidney cancer in terms of epidemiology, histology, hormonal environment, genetics and treatment outcome. This article is written and giving us a sophisticated knowledge to understand the gender difference of renal cell carcinoma.

Please address some queries I pointed out to revise your manuscript.

  1. Line 54; The word “hyperthension” should be replaced to “hypertension”.
  2. Line 67; What is “study3”?
  3. Lines 89-92; Please put a comma between the thousands and hundreds, not a period.
  4. Lines 322-327; Please cite a article for this section.
  5. Lines 344-347; You have mentioned that “the authors concluded that there is no evidence to recommend gender-related bias for immunotherapy, even if the magnitude of efficacy of nivolumab was double for males (12.2 mo of 346 OS improvement for males, 5.8 mo of OS improvement for women)”. I think that the magnitude of efficacy of nivolumab was twice favorable for males, nivolumab should be recommended for males. Please describe the reason more in detail why Graham J et al. did not recommend nivolumab for male patients with RCC.
  6. I want to know the sex difference of the timing of disease diagnosis. I thought that if someone had a less chance of medical check up including ultrasonography or computed tomography, he or she would be more likely diagnosed as RCC in an advanced stage.

Author Response

Dear reviewer,

We have made all the revisions required by the reviewers, using for clarity the software “Track Changes” (Microsoft Word).

The changes are list below: 

  1. We have corrected the typos and wrong word spelling throughout the text.
  2. We did correct the reference incorrectly written as “study3”, as requested, but, later on, we deleted the entire sentence as requested by another reviewer.
  3. We have harmonized the use of dots or commas in numbers (using always dots).
  4. We added the requested references for the paragraph on systemic therapy.
  5. We have clarified the requested parts in the paragraph on systemic therapy. Specifically, the authors reported that the magnitude of efficacy of nivolumab was twice favourable for males than females. Unfortunately, despite the higher efficacy of nivolumab in males, the difference between genders was not statistically strong enough to make clear clinical recommendations based on gender.
  6. We have discussed the impact of gender in the timing of diagnosis, which can be highly influenced by the patients having access to medical check-ups, including ultrasonography or computed tomography.

Reviewer 3 Report

This review article describes the gender issues in renal cell carcinoma. Regardless of the title of the article, this review does not bring anything new to kidney cancer management. The point of reference is still the paper by Lucca, I., Klatte, T., Fajkovic, H. et al. Gender differences in incidence and outcomes of urothelial and kidney cancer. Nat Rev Urol 12, 585–592 (2015). To add, the additional doubt about is lack of appropriate preparation (without appropriate conclusions to moving forward) that is required for a review article intended for the scientific journal.

Author Response

Dear reviewer,

We have added to the reference list the review by Lucca I et al., published in 2015 in Nat Rev Urol. This review includes gender-related topics in bladder cancer, upper urinary tract urothelial carcinoma, and kidney cancer.
Regarding kidney cancer issues, this review talks about incidence, some genetic factors, and outcomes.

In our paper, we have expanded the timelapse, including the last five years, and we have implemented the number of issues, including also surgical data, long-term functional outcomes, and gender-related response to therapy. Moreover, we have also included psychological factors.

We think we have shown that the topic of gender differences in kidney cancer is slowly moving forward, as an ongoing process, which is dynamical, albeit slowly, leading us to the possibility of including genders in clinical nomograms for kidney cancer management.

Round 2

Reviewer 3 Report

The authors have made the necessary changes to the manuscript and introduced corrections improved the scientific soundness.